# Convergent Within-Host Adaptation of *Pseudomonas aeruginosa* through the Transcriptional Regulatory Network

Yair E. Gatt,[a] Dana Savion,[a] Tal Bamberger,[a] Hanah Margalit[a]

[a]Department of Microbiology and Molecular Genetics, Institute for Medical Research Israel-Canada, Faculty of Medicine, The Hebrew University of Jerusalem, Jerusalem, Israel

**ABSTRACT** Bacteria adapt to their host by mutating specific genes and by reprogramming their gene expression. Different strains of a bacterial species often mutate the same genes during infection, demonstrating convergent genetic adaptation. However, there is limited evidence for convergent adaptation at the transcriptional level. To this end, we utilize genomic data of 114 *Pseudomonas aeruginosa* strains, derived from patients with chronic pulmonary infection, and the *P. aeruginosa* transcriptional regulatory network. Relying on loss-of-function mutations in genes encoding transcriptional regulators and predicting their effects through the network, we demonstrate predicted expression changes of the same genes in different strains through different paths in the network, implying convergent transcriptional adaptation. Furthermore, through the transcription lens we associate yet-unknown processes, such as ethanol oxidation and glycine betaine catabolism, with *P. aeruginosa* host adaptation. We also find that known adaptive phenotypes, including antibiotic resistance, which were identified before as achieved by specific mutations, are achieved also through transcriptional changes. Our study has revealed novel interplay between the genetic and transcriptional levels in host adaptation, demonstrating the versatility of the adaptive arsenal of bacterial pathogens and their ability to adapt to the host conditions in a myriad of ways.

**IMPORTANCE** *Pseudomonas aeruginosa* causes significant morbidity and mortality. The pathogen's remarkable ability to establish chronic infections greatly depends on its adaptation to the host environment. Here, we use the transcriptional regulatory network to predict expression changes during adaptation. We expand the processes and functions known to be involved in host adaptation. We show that the pathogen modulates the activity of genes during adaptation, including genes implicated in antibiotic resistance, both directly via genomic mutations and indirectly via mutations in transcriptional regulators. Furthermore, we detect a subgroup of genes whose predicted changes in expression are associated with mucoid strains, a major adaptive phenotype in chronic infections. We propose that these genes constitute the transcriptional arm of the mucoid adaptive strategy. Identification of different adaptive strategies utilized by pathogens during chronic infection has major promise in the treatment of persistent infections and opens the door to personalized tailored antibiotic treatment in the future.

**KEYWORDS** *Pseudomonas aeruginosa*, antibiotic resistance, infectious disease, microbial genetics, transcription regulation network, transcriptional adaptation, within-host adaptation

Address correspondence to Hanah Margalit, hanahm@ekmd.huji.ac.il.

The authors declare no conflict of interest.

Bacterial pathogens utilize diverse strategies for host adaptation during prolonged infections, including changes in the expression of genes (1) and genetic mutations (2). While changes at the transcriptional level have traditionally been considered within the setting of acute infection, their strength in characterizing chronic isolates and key

role in chronic adaptation have come into prominence in recent years with the emergence and availability of high-throughput RNA sequencing methodologies (3–5). Likewise, advances in DNA sequencing have allowed the detection of mutations accumulating during chronic infections (6). Previous large-scale systematic studies have shown that different strains of a bacterial species, infecting different patients, mutate the same genes during infection, indicating within-host convergent adaptation at the genetic level (7, 8). Research of convergent adaptation at the transcriptional level was so far limited to small numbers of clinical strains. Laboratory evolution experiments using two parallel strains of *Escherichia coli*, derived from the same ancestor strain, identified similar expression changes in a subgroup of genes, suggesting convergent adaptation at the transcriptional level (9). Likewise, convergent transcriptional adaptation has been demonstrated for subpopulations of *Pseudomonas aeruginosa* infecting the same lung (4) or different sites (10) and for very small numbers of strains during infection (3, 11–13). Yet, lack of large-scale transcriptomic data for multiple strains hampered systematic research addressing the convergence of transcriptional adaptation between different strains infecting different patients. Here, we address this challenge by exploiting the transcriptional regulatory network of a bacterial pathogen and ample longitudinal data of genomic sequences derived from patients during chronic infections, using *P. aeruginosa* as a model.

*P. aeruginosa* is an opportunistic bacterial pathogen causing significant morbidity and mortality worldwide (14). *P. aeruginosa* has a variety of metabolic and pathogenic abilities, allowing it to infect and colonize a vast range of hosts (15). It is the predominant pathogen associated with cystic fibrosis (CF) pulmonary infections, with clinical presentations ranging from acute pneumonia lasting days to chronic infections lasting dozens of years (16). Managing these infections is a major challenge in the treatment of patients with CF, as *P. aeruginosa* strains develop a broad spectrum of tolerance and later resistance to antibiotic treatments during their prolonged infections (17–19). Isolated colonies with a mucoid phenotype are a hallmark of prolonged *P. aeruginosa* infection (2). These colonies are defined by the overproduction of alginate, and over 90% of them are associated with loss-of-function (LOF) mutations in the *mucA* gene, encoding an anti-sigma factor preventing the activity of AlgU, the master regulator of alginate production (20).

Selective pressures acting on pathogens during chronic infections are commonly reflected in genome degradation and LOF mutations (21). In a previous study, we analyzed published genomic data from *P. aeruginosa* longitudinal isolates derived from 114 chronically infected patients (see Table S1 in the supplemental material) and showed that the same bacterial genes undergo mutations during infection by different strains (7). Genes identified to repeatedly undergo LOF during infection in a statistically significant number of patients were considered indicators of adaptation. LOF mutations in 59 genes were identified as involved in host adaptation, while additional LOF mutations were identified among small groups of patients. Interestingly, many of the genes undergoing LOF are related to transcription regulation, suggesting that the LOF of these genes may affect additional genes, targeted by those regulators. Such secondary transcriptional effects due to different regulators in the different strains could converge on common targets, hinting at convergent adaptation also at the transcriptional level (Fig. 1).

Convergent adaptation at the transcriptional level can be systematically studied by predicting the effect of mutations through the transcriptional regulatory network (TRN). In this network, the transcription factors (TFs), sigma factors (SFs), and genes are represented as nodes, and directed edges represent regulatory interactions between TFs/SFs and their target genes. Since the LOF of TFs/SFs affect genes that are downstream of them in the TRN, analyzing the propagation of these effects can demonstrate long-range impacts of genomic changes. It is possible that different LOF events in the network have similar long-range effects, converging at the same genes or pathways. Analysis of the long-range effects in the TRN may therefore enable the detection of convergent adaptation at the transcriptional level that was undetectable in previous

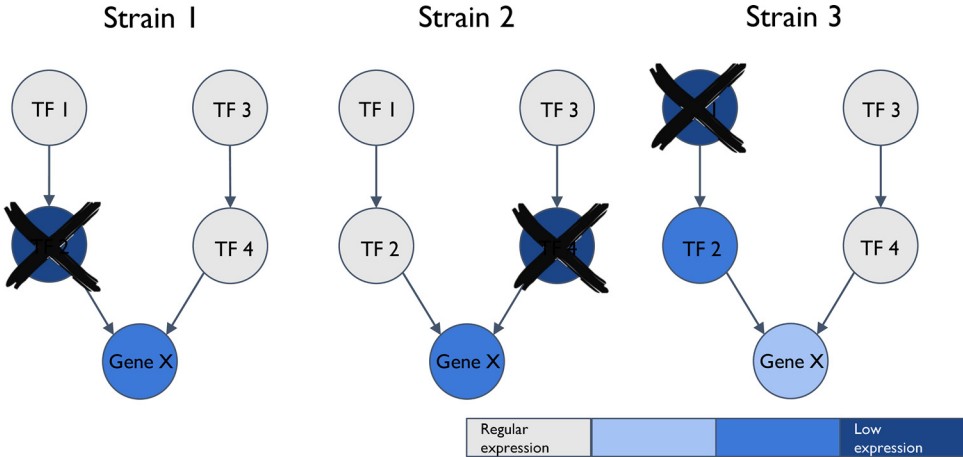

**FIG 1** Different genetic events can converge at the transcriptional level. Gene X is positively regulated by TF2 and TF4, TF2 is positively regulated by TF1, and TF4 is positively regulated by TF3. Three different LOF events occurred in the three different strains. TF2 underwent LOF in strain 1, TF4 underwent LOF in strain 2, and TF1 underwent LOF in strain 3. Despite these differences, all three events are expected to lead to some reduction in the expression of gene X, demonstrating transcriptional convergence. Note that all analyses in this study were performed at the level of progenitor-progeny isolate pairs of the strains.

analyses (8). Multiple studies of the *P. aeruginosa* TRN have recently been published, utilizing various computational (22–24) and experimental (15, 25) methodologies.

Here, we integrate our LOF data (7) within the *P. aeruginosa* experimental TRN (15, 25) and follow the propagated effects of LOF of regulators through the TRN for each of the strains in our data. Remarkably, we identify genes that are consistently predicted to change in their expression in an identical manner in different strains due to different genetic events and through different paths in the TRN, implying convergent adaptation at the transcriptional level of different strains. Our results expand both the arsenal of strategies employed by the pathogen for host adaptation and the repertoire of genes involved in this process.

## RESULTS

**Multiple genes show convergent adaptation at the transcriptional level.** To follow changes in gene expression due to LOF mutations in genes encoding transcriptional regulators, we need to determine genomic changes during infection and to have data on transcriptional regulatory interactions in *P. aeruginosa*. The former were obtained from our previous study, in which we analyzed genomic sequences of *P. aeruginosa* longitudinal isolates derived from 114 chronically infected patients (see Table S1 in the supplemental material for a list of the patients and available clinical details), where samples were isolated from each patient at least at two time points during the infection (7). All isolates derived from a single patient were defined as a "strain." We developed an algorithm called TRACE to determine the phylogenetic relationships between the isolates of each strain. This algorithm determines high-confidence progenitor-progeny pairs among the isolates of each strain (Text S1). In such pairs one isolate is highly likely to have derived from the other isolate during the infection. This resulted in 260 progenitor-progeny isolate pairs derived for the 114 strains in our data. All further analyses in this paper are conducted using these 260 pairs. The time interval between the sampling of the progenitor isolate and progeny isolate across all pairs was highly variable, ranging from days to dozens of years (Fig. S1a). Genomic changes between the sampling of progenitor and progeny isolates in these pairs are highly likely to have occurred during the infection. We determined genes undergoing LOF between isolates in each such pair using a pipeline based on the *breseq* and snpEff algorithms (26, 27), including genes encoding transcriptional regulators (TFs and SFs, Fig. S1b).

We constructed the TRN of *P. aeruginosa* PAO1 using experimental transcription regulation data compiled in the study by Galán-Vásquez et al. (25) (Materials and

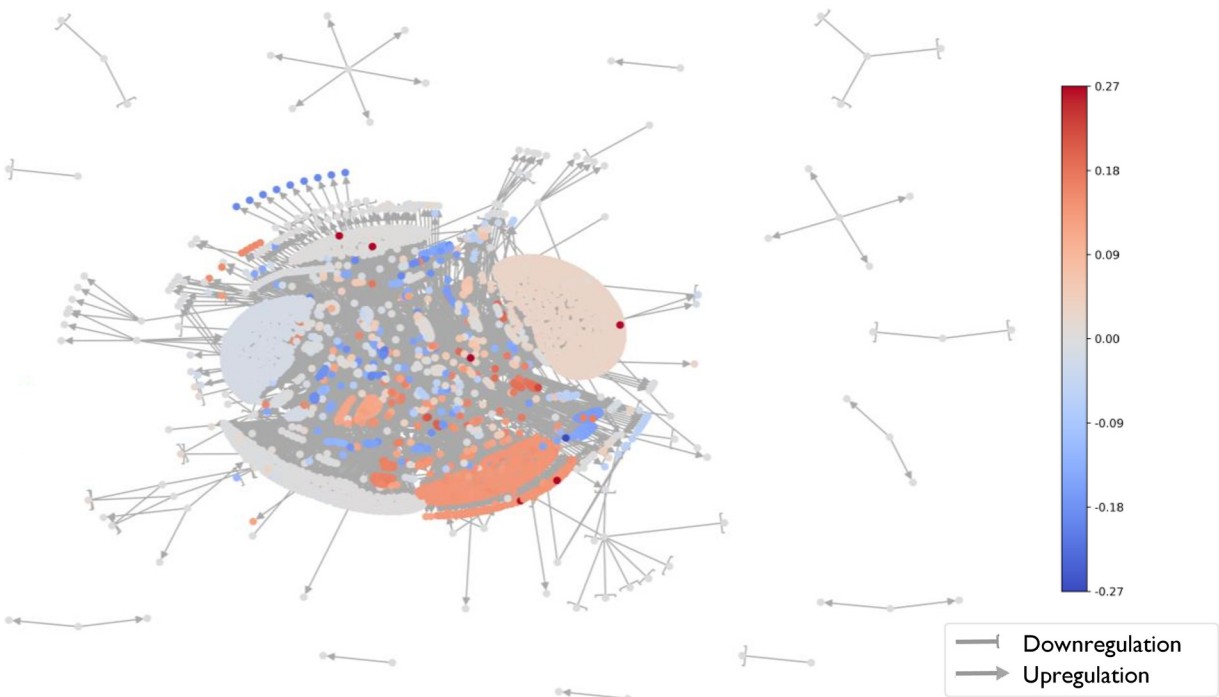

**FIG 2** Mean regulatory outcome scores (mROS) of genes across the transcriptional regulatory network of *P. aeruginosa*. The transcriptional regulatory network of *P. aeruginosa* (25) is shown, with nodes colored by the mROS values of the corresponding genes. Red, positive mROS; blue, negative mROS; arrows, positive regulation; blunt-end arrows, negative regulation.

Methods). This TRN contains 2,831 nodes, encompassing 50.8% of the total number of genes in *P. aeruginosa* PAO1. One hundred sixty-three of the nodes represent TFs or SFs, and 2,668 represent target genes. The TRN has 4,827 directed edges, which correspond to up- or downregulation of the target gene by the transcriptional regulator, depending on its regulation type. Thirty-eight percent of TF/SF-encoding genes included in the network were identified as undergoing LOF in at least one progenitor-progeny pair, with 18% of all TF/SF-encoding genes undergoing LOF in two or more pairs.

The LOF of TF/SF-encoding genes can have long-range effects in the TRN by affecting the expression of the targets of those regulators. Targets themselves encoding TF/SFs will affect the expression of their own targets and so forth, propagating the effect through the network. We generated a TRN for each progenitor-progeny isolate pair and developed a propagation algorithm (Materials and Methods) to predict the effects of TFs/SFs undergoing LOF between the progenitor and progeny isolates. To assess the impact of TF/SF-encoding genes undergoing LOF on downstream genes in the network, we considered the shortest path between the regulator and downstream gene and the type of regulation (positive/negative), assuming additive effects of TFs/SFs regulating the same targets. This computation results in a ternary regulatory outcome score (ROS) of $-1/+1/0$ for each node, representing a predicted decrease, increase, or no change in the expected expression of the gene between the progenitor and progeny isolates, respectively. For all genes, the value of ROS across the different progenitor-progeny pairs was not associated with the time interval between the sampling of the progenitor and that of the progeny isolates (Materials and Methods), recapitulating that most genomic changes occur at the early stages of the infection (7, 13, 28).

Next, we generated a mean scored TRN, where the score of each gene is the mean of its ROS values (mROS) across all the progenitor-progeny pairs (Fig. 2; Table S2). A relatively high positive mROS indicates that the expression of a gene is predicted to consistently increase across different progenitor-progeny pairs during infection, whereas a relatively low negative mROS indicates that the expression is predicted to consistently decrease during infection. The scores of different genes range from $-0.21$ to $+0.27$

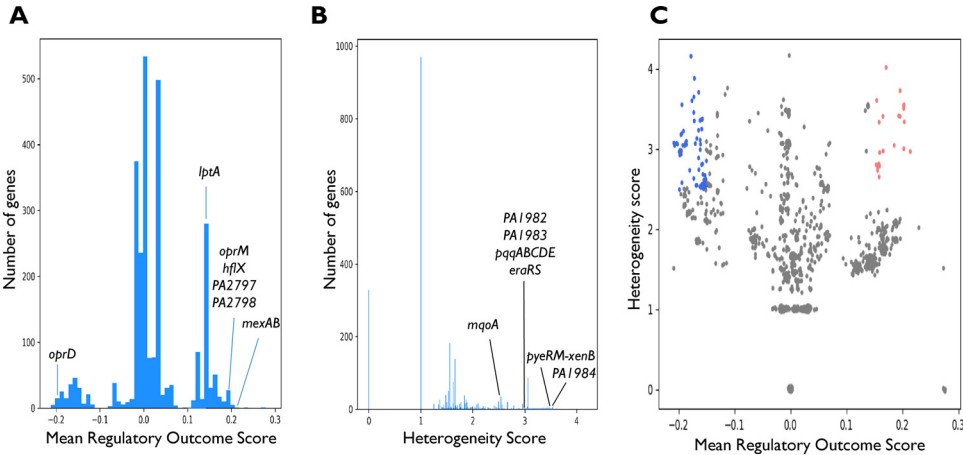

**FIG 3** Distribution of mean regulatory outcome scores (mROS values) and heterogeneity scores (HS values). (A and B) mROS (A) and HS (B) values of all genes in the transcriptional regulatory network. Notable genes mentioned in the main text are denoted. (C) Plot of mROS versus HS values of all genes. Genes with mROS values of >0.15 and HS values of >2.5 are marked in red, and those with mROS values of <−0.15 and HS values of >2.5 are marked in blue.

(Fig. 3A). The same value of mROS can be achieved in different manners, i.e., an mROS of 0.2 could be achieved by the gene predicted to be upregulated in 0.2 of the pairs, with no predicted change in the rest of the pairs, or by being upregulated in 0.6 and downregulated in 0.4 of the pairs. We therefore calculated for each gene the fraction of progenitor-progeny pairs where the gene is predicted to change in the same direction as the sign of the mROS (Table S2, Fraction of pairs with major outcome). The maximum fraction of progenitor-progeny pairs showing predicted change in the same direction as mROS was 0.3. While this fraction seems modest, it is important to emphasize that it includes many progenitor-progeny pairs in which relevant regulators do not undergo LOF. In those progenitor-progeny pairs in which upstream TFs/SFs undergo LOF, the directions of expression change in some downstream targets are highly consistent among the various progenitor-progeny pairs, with genes with |mROS| values of >0.15 having the same ROS (−1 or 1) in 0.76 to 0.96 of the pairs (Table S2, Major outcome out of nonzeros). In addition, to evaluate the mROS results and avoid biases due to the structure of the network, we developed a statistical method to assess whether the mROS of a gene statistically significantly differs from random expectation (Materials and Methods). Approximately 2,000 genes have statistically significant P values, indicating that even |mROS| values of 0.05 do not occur at random due to the structure of the network and are the consequence of the specific genes undergoing LOF. Still, we consider only genes with |mROS| values above 0.15 that are statistically significant for further discussion.

Negative mROS values imply predicted downregulation of the gene and, hence, reduced expression of its protein product. However, the protein expression of a gene could also be decreased by LOF of the gene itself. For each gene, we calculated the fraction of the progenitor-progeny pairs where the gene underwent LOF itself out of all the pairs where the gene either underwent LOF or obtained an ROS of −1. Our results show that the decreases in expression of all genes with low mROS values (<−0.15) are related to upstream events in the great majority of the progenitor-progeny pairs rather than to LOF of the gene itself (Table S2, Contribution of LOF). This also applies to genes previously shown to undergo adaptive LOF, such as *oprD* and *PA2403* (7), implying that the TRN has a major impact on tailoring also the expression of genes undergoing adaptive LOF themselves.

A repeatedly predicted change in the expression of a gene in different strains might occur due to the same genetic event, i.e., LOF of the same TF/SF, repeating in different strains, or due to transcriptional convergence of different genetic events through the TRN. The first case suggests selection at the genetic level for the specific genetic event

causing the change, whereas the second case suggests selection at the transcriptional level for the change in expression itself. To study whether the same regulatory outcome is achieved through various paths in the network, we defined a heterogeneity score (HS) for each gene. The HS assesses how many LOF events of upstream TFs/SFs determine a gene's ROS in the different progenitor-progeny pairs, while considering the relative impact of each LOF event on the mROS (Materials and Methods). The HS values range from 0.0 to 4.2 (Fig. 3B), with most genes being assigned a score of 1.0, meaning their mROS was affected by LOF events of only a single node. Based on the HS distribution (Fig. 3B), we determined genes with an HS of >2.5 as having a high HS (compared to the rest of the genome). Three hundred sixteen genes have a high HS, indicating that their mROS is determined by different events in the various progenitor-progeny pairs and strongly supporting selection at the transcriptional level. We detect 118 genes that have both high HS and extreme mROS values, indicating convergent adaptation of these genes at the transcriptional level (Fig. 3C; Table S2). A detailed analysis of these genes can be found in Text S1.

**Known antibiotic resistance genes are predicted to consistently change their expression across different progenitor-progeny pairs.** Several genes related to antibiotic resistance obtain statistically significant high or low mROS values, suggesting their up- or downregulation, respectively. *oprD* has an extremely low mROS (mROS, −0.19; fraction of pairs with major outcome, 0.24), consistent with increased carbapenem resistance known to be associated with its LOF (29). This is also in line with previous high-throughput studies that found reduced *oprD* expression to be an indicator for carbapenem resistance (30, 31). We identify both progenitor-progeny pairs where *oprD* undergoes LOF, also noted in our previous study (7), and pairs where its expression is predicted to decrease via the LOF of TFs regulating it. Most notably, we predict a decrease in the expression of *oprD* in a strain previously noted to develop carbapenem resistance with no known genetic correlate (32), supporting the clinical relevance of this mechanism. The *mexAB-oprM* and *mexXY* genes, which encode major multidrug efflux pumps in *P. aeruginosa* (33), all have very high mROS values (mROS, 0.17 to 0.21; fraction of pairs with major outcome, 0.23 to 0.25), predicting an increase in their expression, which is consistent with increased antibiotic resistance. These genes also have high HS values (3.0 to 3.1 for *mexAB-oprM*), indicating selection for their increased expression at the transcriptional level. Here, too, we predict reduced expression of *mexR* and increased expression of the *mexAB-oprM* efflux pump in a strain that developed aztreonam resistance during infection with no known genetic correlate, providing an explanation for the resistant phenotype (34).

Most *P. aeruginosa* clinical strains are exposed to aminoglycoside antibiotics, and indeed genes that have been linked to aminoglycoside resistance are among the genes with the highest mROS values (mROS, 0.14 to 0.19; fraction of pairs with major outcome, 0.20 to 0.22). These include the *PA2797-PA2798* two-component system (TCS), where activating mutations have been linked to resistance (35), *hflX*, whose increased copy number has been linked to resistance (36), and *lptA*, whose deletion has been linked to increased sensitivity to aminoglycosides (35). To verify our predictions regarding antibiotic resistance, we searched for strains that were shown to have changes in the MIC for different antibiotics, which could not be explained in the original studies by any genetic changes. In addition to the two cases noted above, we found several cases in two different studies (32, 34), where our predicted changes in expression through the transcriptional regulatory network could sensibly explain the changes in MIC (Text S1). Together, these results suggest that in addition to well-established genetic antibiotic resistance mechanisms, different *P. aeruginosa* strains also tailor the expression levels of genes through the transcriptional regulatory network to achieve antibiotic resistance during infection.

Many other genes with extreme mROS values are in line with known adaptive processes in *P. aeruginosa*, with consistent predicted upregulation of genes related to biofilm across the different progenitor-progeny pairs and downregulation of genes related to motility and siderophore biosynthesis (Table S2).

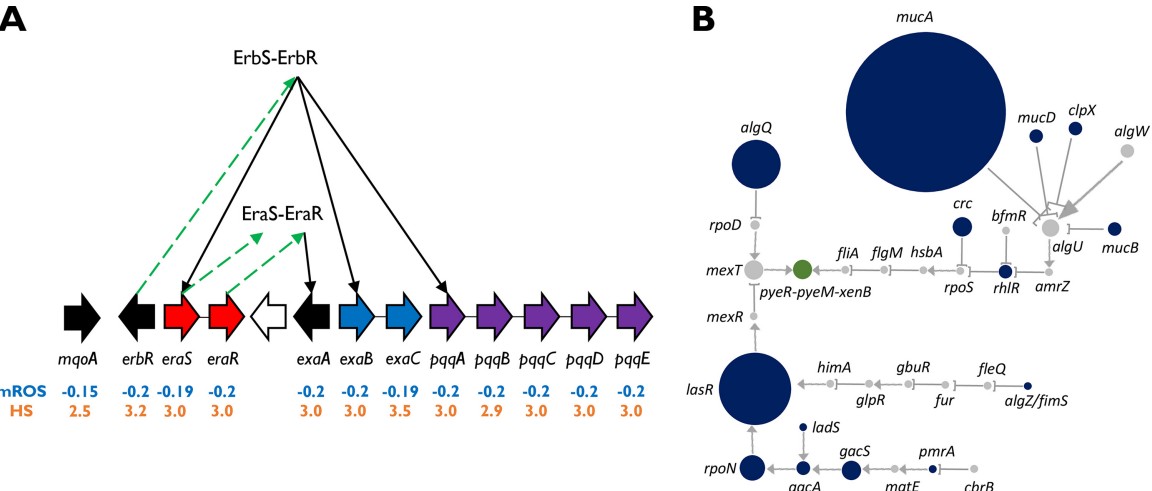

**FIG 4** Heterogeneity score (HS) associates novel processes and genes with host adaptation. (A) mROS and HS values of the components of the ethanol oxidation system. The ErbSR TCS regulates most components involved in ethanol oxidation, including the EraSR TCS, which regulates *exaA*. For clarity, additional regulators are not shown. All genes involved in this system have low mROS values (blue) and high HS values (orange). Wide arrows show genes in their corresponding genomic locations; all genes reside in the same genomic region except *mqoA*, which is in a different locus. Arrows pointing to the right correspond to genes on the forward strand, and arrows pointing to the left correspond to genes on the reverse strand. Same-operon genes are in the same color. Green dashed arrows connect genes to their products. Black arrows correspond to positive regulation by transcription factors. (B) LOF events of TFs/SFs leading to predicted increased expression of the *pyeR-pyeM-xenB* operon in the different strains. Arrows, positive regulation; blunt-end arrows, negative regulation; green node, *pyeR-pyeM-xenB*; blue nodes, nodes of genes encoding TFs/SFs whose LOFs lead to predicted increased expression of *pyeR-pyeM-xenB*; gray nodes, all other nodes. The size of a node is proportional to the number of progenitor-progeny pairs in which it underwent LOF.

**The expression of genes related to ethanol oxidation is predicted to consistently decrease in different progenitor-progeny pairs due to different genetic events.** Remarkably, one of the operons whose genes have the lowest mROS values, alongside high HS values, is the pyrroloquinoline quinone (PQQ) biosynthesis operon *pqqABCDE* (mROS, −0.2; fraction of pairs with major outcome, 0.23 to 0.24; HS, 2.9 to 3.0). *P. aeruginosa* uses the pyrroloquinoline quinone (PQQ)-dependent ethanol oxidation system when growing on ethanol (37). This PQQ-dependent system comprises a quinoprotein ethanol dehydrogenase (PA1982), cytochrome C550 (PA1983), and an NAD-dependent acetaldehyde dehydrogenase (PA1984). The genes encoding all these components have high HS and low mROS values (Fig. 4A; Table S2) (mROS, −0.2 to −0.19; fractions of pairs with major outcome, 0.23 to 0.24; HS, 2.9 to 3.5). *mqoA*, which is also required for this system, also has a low mROS (mROS, −0.15; fraction of pairs with major outcome, 0.2) and a high HS (2.5). Ethanol dehydrogenase itself is regulated by the EraRS TCS (37) and the ErbRS TCS (38), and the genes encoding the components of these TCSs also have low mROS values with high HS values (Fig. 4A; Table S2) (mROS, −0.2 to −0.19; fractions of pairs with major outcome, 0.23 to 0.24; HS, 3.0 to 3.2). Overall, these results show that reduced expression of the ethanol oxidation system is strongly selected for across different strains of *P. aeruginosa* during infection, with many different events converging to downregulate various genes related to this system across different strains. The strong negative selection can possibly be explained by the scarcity of ethanol in the CF lung environment compared to external environments from which infecting *P. aeruginosa* strains might derive and the high energetic cost of constantly producing the extensive ethanol oxidation machinery.

An example of convergence to increased expression through the TRN (high mROS and high HS) is the *pyeR-pyeM-xenB* operon (*PA4354-PA4355-xenB*), with an mROS of 0.2, a fraction of pairs with major outcome of 0.24, and an HS of 3.5. This operon is predicted to be upregulated in a large fraction of the progenitor-progeny pairs due to variable LOF events of upstream TFs/SFs (Fig. 4B). While the PyeR TF has been linked to biofilm formation, the roles of PyeM and XenB, as well as the cooperation between these three components, are still not clear (39). The high HS of this operon indicates strong selection for

its upregulation during infection and provides strong motivation for further elucidation of its function and the interplay between its different components.

**Analysis of metacategories associates novel processes with host adaptation.** In our previous study, we found that different bacterial species undergoing within-host adaptation accumulate changes in different genes belonging to the same KEGG pathways and relating to the same Gene Ontology (GO) annotations (7). KEGG is a database that links genomic information with higher-order functional information of pathways (40). Similarly, GO annotations involve metacategories characterizing genes by the functional, spatial, and structural information of their products (41). To identify convergence through the transcriptional regulatory network that relates to such metacategories, we compared the mROS and HS values of all genes corresponding to each KEGG pathway or GO annotation to the scores of the rest of the genes. We performed a two-sided or one-sided Mann-Whitney test for identifying statistically significant differences in the mROS and HS, respectively, and corrected the *P* value*s* for the testing of multiple hypotheses using the Benjamini-Hochberg method (42). The genes in 11 pathways have statistically significantly higher or lower mROS values than the remaining genes, and the genes in five pathways have higher HS values, including three pathways where both the mROS and HS differ statistically significantly from the corresponding values of the rest of the genes (Fig. 5, Fig. S2, and Table S3). Most of these pathways, such as flagellum and siderophore biosynthesis (average mROS, −0.10 to −0.05; average fraction of pairs with major outcome, 0.15), were identified in our previous study as related to host adaptation of *P. aeruginosa* because genes included in them accumulated mutations (7). Here, we show that in addition to the direct mutations, many of these pathways are also affected during infection through mutations in various transcription factors that are predicted to affect the expression of genes in the pathways.

The polyketide sugar unit biosynthesis (PSUB) pathway genes have statistically significant low mROS values compared with other genes, meaning that genes in this pathway are predicted to be downregulated (average mROS, −0.12; average fraction of pairs with major outcome, 0.17). This pathway is related to the synthesis of ʟ-rhamnose, a component of the highly immunogenic *P. aeruginosa* lipopolysaccharide (LPS) and other antigens (43). We previously demonstrated that mutations in LPS biosynthesis are common during host adaptation of different organisms, which we believe is related to immune evasion. Conceivably, the predicted reduced expression achieves a similar effect or alternatively could prevent the accumulation of possibly toxic ʟ-rhamnose components under impaired synthesis conditions. If the second explanation is correct, we would expect both mutations in LPS biosynthesis genes and predicted downregulation of PSUB genes to occur in the same strains. In fact, only eight strains both accumulated mutations in LPS and were predicted to have PSUB downregulation during the infection, representing 36.3% of the first group and 28.6% of the second group ($P = 0.12$ by hypergeometric test). The independent occurrence of these two phenomena in most strains supports the first interpretation.

Similarly, 47 GO annotations have statistically significant high or low mROS values, and 23 have statistically significant high HS values, including 12 where both measures are statistically significant (Fig. S2; Table S3). Many of those annotations are related to known adaptive processes such as chemotaxis and alginate and polysaccharide biosynthesis. Some annotations correspond to processes not previously linked to host adaptation. Most notably, the expression of genes related to glycine betaine catabolism is predicted to greatly increase across different progenitor-progeny pairs with statistically significant HS values (average HS, 1.9; average mROS, 0.11; average fraction of pairs with major outcome, 0.18). Glycine betaine is the major osmolyte in *P. aeruginosa*, and changes in the glycine betaine pool are a hallmark of *P. aeruginosa* adaptation to different osmotic conditions. An increase in glycine betaine catabolism causes reduced levels of this osmolyte. Low glycine betaine levels have indeed been demonstrated in metabolic studies of host-adapted *P. aeruginosa* strains and shown to lead to increased susceptibility to hyperosmotic stress (44). This mechanism is hypothesized to underlie the proven benefit of hyperosmotic saline therapy in CF patients (45).

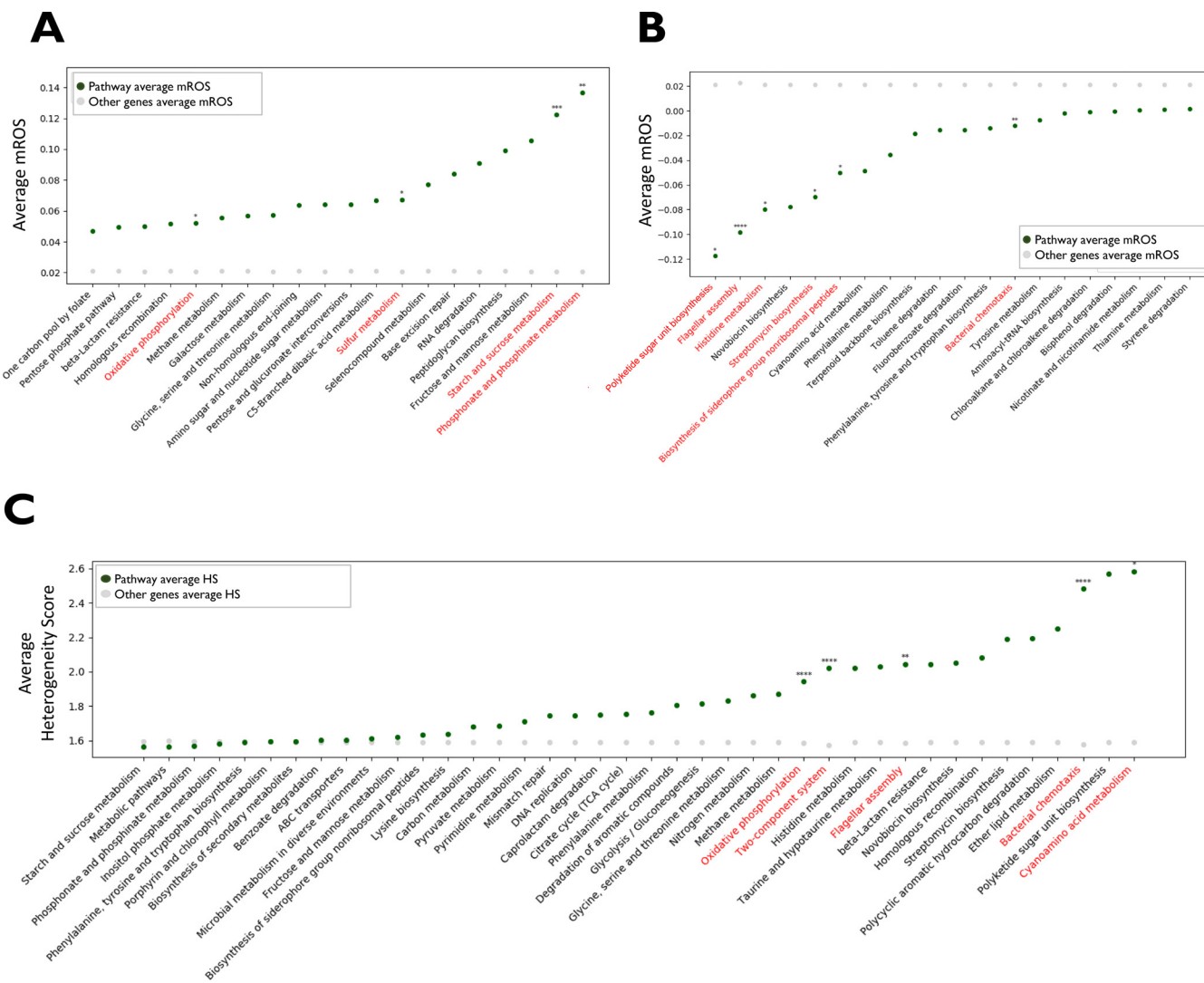

**FIG 5** Mean regulatory outcome scores (mROS) and heterogeneity scores (HS) of selected KEGG pathways. Selected KEGG pathways with the highest average mROS values (A), the lowest average mROS values (B), and the highest average HS values (C). Green dots indicate the mean value for the genes included in the pathway, and gray dots indicate the mean for all genes not included in the pathway. Pathways passing the statistical test for the mROS or HS values are written in red. *, $P$ value < 0.1; **, $P$ value < 0.01; ***, $P$ value < 0.001; ****, $P$ value <0.0001. All $P$ values are corrected for the testing of multiple hypotheses. TCA, tricarboxylic acid.

In addition to LOF mutations, in our previous study (7) we also collected information on other nonsynonymous mutations. We defined KEGG pathways and GO annotations where the different genes accumulated nonsynonymous mutations in a statistically significant number of strains as adaptive (7). Intriguingly, we note a large overlap between these pathways and annotations and pathways and annotations with extreme mROS and high HS values. Pathways and annotations defined as adaptive in our previous study (7) have lower mROS and higher HS values than other pathways (Fig. S3), indicating strong selection for the decreased expression of genes in these pathways. Adaptive GO annotations are also enriched with GO annotations with statistically significant extreme mROS and HS values ($P = 0.027$ for HS and $P = 0.0006$ for mROS, by Fisher's exact test). This demonstrates the interplay between adaptation at the genetic and transcriptional levels and the diverse means by which the pathogen can modulate the same processes.

**Clustering analysis demonstrates a unique transcriptional response associated with mucoid strains.** In the previous sections, we demonstrated that the expression of some genes was predicted to change in a consistent direction across different progenitor-

progeny pairs (e.g., *oprD* and *mexAB*). Nevertheless, no gene had an mROS of >0.3 or <−0.3, suggesting that these results are obtained due to consistent adaptive strategies in subsets of progenitor-progeny pairs. These may relate to specific selective pressures pertinent to distinct subsets of progenitor-progeny pairs, such as different antibiotic treatments. Furthermore, a gene that is predicted to change in expression during adaptation does not operate in a void but as part of an adaptive strategy involving changes in multiple genes and pathways. Clustering analysis allows grouping of genes predicted to change in the same progenitor-progeny pairs to "clusters." This would allow the detection of genes involved in the same adaptation strategy and the determination of subsets of progenitor-progeny pairs where these genes show consistent predicted expression changes. To this end, we defined a vector for each gene consisting of its ROS values in the different progenitor-progeny pairs, resulting in a vector of 0s, 1s, and −1s for each gene. We then calculated the Spearman correlation coefficient ($r$) between each two vectors. Finally, we clustered the different genes based on the vectors and $1 - |r|$, used as the distance between two genes (Materials and Methods). We used the absolute value of the correlation coefficients since the expression of genes acting as part of the same adaptive strategy may change in the same or opposite directions (e.g., reduced motility and increased biofilm production).

This analysis resulted in a total of 14 gene clusters, including seven small clusters (<10 genes), three medium clusters (11 to 70 genes), and four large clusters (300 to 800 genes) (Fig. S4; Table S4). Genes that are found in the same cluster may simply be coregulated by the same transcriptional regulator. Such cases are not likely to represent independent LOF events of TFs/SFs subject to the same selective pressures. To filter out these cases, we searched for clusters including independent pairs of genes—genes that do not have a shared transcriptional regulator (ancestor) in a distance less than four edges from the two genes in the network (Text S1).

One of the large clusters (cluster 14) includes thousands of independent correlated gene pairs. These are derived from several independent groups of genes in the TRN (Fig. S4). The coordinated changes in the expression of hundreds of genes, which are not coregulated, suggest that this cluster corresponds to an adaptive strategy employing transcriptome-wide changes. To further decipher the functional correlates of this cluster, we studied the functional categories to which the different genes in the cluster belong. In this analysis, we included, in addition to KEGG pathways and GO annotations, pathways described in PseudoCyc, a database that includes descriptions of metabolic processes in *P. aeruginosa* and the genes involved in them (46). We assigned genes in the cluster to their corresponding KEGG pathways, GO annotations, and PseudoCyc pathways and summed the total mROS values of each pathway for the cluster (Table S4). Pathways or annotations with a negative total mROS are expected to decrease in activity, whereas pathways or annotations with a positive total mROS are expected to increase.

Many known pathways associated with host adaptation of *P. aeruginosa* change in accordance with their known role in adaptation in cluster 14. This includes an increase in alginate production, anaerobic respiration, fatty acid metabolism (47), degradation of superoxide radicals and sulfate assimilation (5), and a decrease in O-antigen biosynthesis and siderophore biosynthesis. These findings suggest that cluster 14 may be associated with the well-described mucoid adaptation of *P. aeruginosa*. The cluster also includes pathways previously not associated with host adaptation, including peptidoglycan biosynthesis and the ethanol oxidation system mentioned above. The cluster is also associated with various changes in the metabolism of amino acids and central metabolism (Text S1).

We attempted to assess whether the predicted changes of genes in the cluster are associated with specific subsets of progenitor-progeny pairs. For this purpose, we developed a concordance score, measuring whether the predicted changes of each progenitor-progeny pair are in line with the mROS of the genes in the cluster. Briefly, we compared the ROS of each gene within each progenitor-progeny pair with the gene's mROS. If the ROS has the same sign as the mROS, the gene in the specific

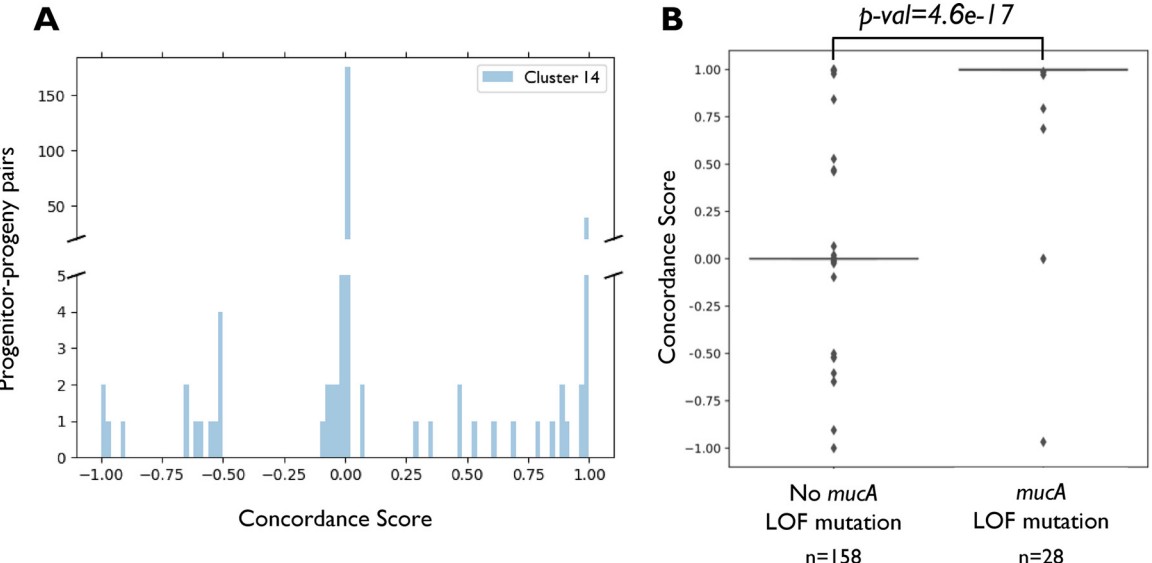

**FIG 6** The concordance score distinguishes a subset of strains associated with mucoid adaptation. (A) Histogram of the concordance scores of progenitor-progeny pairs for cluster 14. (B) Box plots of the concordance scores for cluster 14 of progenitor-progeny pairs with and without *mucA* LOF mutations.

progenitor-progeny pair obtained a score of $+1$; if it has the opposite sign, it obtained a score of $-1$; and if the ROS has the value of 0, the gene obtained a score of 0. We then averaged the scores of all genes in the cluster for each progenitor-progeny pair, obtaining a concordance score for the cluster for that pair. The concordance score for cluster 14 has a bimodal distribution across the different progenitor-progeny pairs, with a larger peak around a score of 0 and a smaller peak around 1 (Fig. 6A). Hence, there is a subset of progenitor-progeny pairs where the expression of close to 100% of genes in cluster 14 is predicted to change according to their mROS. This has allowed us to associate 46 progenitor-progeny pairs with a concordance score of $>0.75$ with cluster 14. These progenitor-progeny pairs were derived from 26 patients reported in 10 different studies (8, 32, 34, 48–54). When clustering the progenitor-progeny pairs based on the ROS vectors across genes, we can see the progenitor-progeny pairs associated with cluster 14 are indeed strongly clustered together, further supporting the association of the changes in this group of genes with a specific subset of progenitor-progeny pairs (Fig. S5).

The association of cluster 14 with alginate production and other well-described adaptive qualities suggests its relation to mucoid adaptation. Indeed, progenitor-progeny pairs with LOF mutations in *mucA* had statistically significantly far-higher concordance scores for cluster 14 than other progenitor-progeny pairs (Fig. 6B). These results indicate that cluster 14 is well correlated with mucoid isolates of *P. aeruginosa* and constitutes a set of coherent transcriptomic changes occurring in these isolates.

## DISCUSSION

**Predicted convergent transcriptional adaptation can be discovered through the transcriptional regulatory network.** In this work, we studied within-host convergent adaptation through the transcriptional regulatory network of *P. aeruginosa* during chronic infection of 114 patients with pulmonary disease. We developed a novel propagation algorithm to predict the impact of LOF mutations in TF/SF-encoding genes on the expression of other genes in the TRN during the infection of different strains. This algorithm has enabled the detection of genes with consistent predicted expression change during the infection across different strains, suggesting convergent adaptation at the transcriptional level.

The nature of the propagation algorithm is quite straightforward, as it considers only the progenitor-progeny pairs in which a gene underwent LOF and the location of

the gene within the TRN. We do not delve into predicting the exact magnitude of the change in expression or into predicting changes in expression related to missense mutations. Our nonambiguous goal of only assigning each gene a ternary score in each progenitor-progeny pair allows us to make predictions with relative confidence and minimal assumptions. Our focus on genes that repeatedly increase or decrease in expression in different strains also makes our algorithm robust to rare events that may bias predictions in specific strains.

We examined whether the consistent predicted changes in the expression of genes are derived from the same genetic events across different progenitor-progeny pairs or from different genetic events. The former case has been studied extensively previously (7) and indicates selection at the genetic level. In contrast, the latter indicates selection at the transcriptional level, with adaptive trajectories of different strains undergoing different genetic events converging at the transcriptional level. This measure is expressed by the heterogeneity score (HS) we developed, and we indeed find clear selection for transcriptional convergence for a large number of genes. There are several caveats associated with our results. The first one concerns the size of the TRN. The network was compiled by Galán-Vásquez et al. (25) from previously published assorted experimental data and contains 50% of the genes of *P. aeruginosa*, likely missing regulatory interactions. Lack of genes and network connections may bias predictions and provide only partial results. Nevertheless, it is still by far the most extensive experimental TRN published for *P. aeruginosa* to date. Additionally, the "minimal frustration" property of biological networks has recently been demonstrated to allow accurate predictions to be made based on incomplete networks (55, 56). The second caveat regards the assumption of additive effects of the different regulators. This assumption may not be appropriate for some genes with more complex regulatory mechanisms. Finally, we also assume baseline expression of all genes in the TRN under the infecting conditions, whereas empirical evidence demonstrates that the environmental and intracellular conditions have a great impact on the subgroup of genes that is expressed (5). Since only a portion of the genome is expressed under specific conditions, our assumption may lead to propagation of the effect of LOF mutations in TF/SF-encoding genes via genes that are not in fact expressed under the studied conditions. Nevertheless, the changes in expression that we predict occur due to heterogeneous genetic events, suggesting that at least some of the genes in the paths we identify may be expressed and the effect of LOF of TF/SF-encoding genes can propagate through them. Previous studies have shown the transcriptional profile of *P. aeruginosa* to be better linked to the environment than to the genetic profile (4, 5, 57). Consistent with these studies and others (58), our results suggest that this property may be achieved by different genetic events leading to the same transcriptional outcomes.

Convergence at the transcriptional level is a novel concept in host adaptation and harbors opportunity for detecting yet-unknown adaptive processes. Phenotype convergence in adaptation to the host may be more widespread than we predict, as even different transcriptional responses have been shown to converge to the same phenotype or similar phenotypes (10, 59). We find that *P. aeruginosa* utilizes LOF of TFs and SFs to tailor the expression of genes known to be involved in antibiotic resistance, without accumulating mutations directly in those genes or their promoters. Sequencing-based methods have recently been gaining popularity for the rapid detection of antibiotic resistance without the need to culture infecting pathogens (60). Our results indicate that such methods must consider the ability of *P. aeruginosa* to achieve antibiotic persistence or resistance via the propagation of effects through the TRN, without mutations in known genes associated with resistance phenotypes. In addition, our results implicate selection against ethanol oxidation as a major adaptive process during *P. aeruginosa* chronic infection. To our knowledge, this is the first time the downregulation of this process has been linked to host adaptation, and it may open the door to new ethanol-based therapeutic approaches in CF, such as ethanol inhalation, which has recently been studied in relation to COVID-19 (61). Notably, the straightforward methods we developed can be applied to additional bacterial species annotated in our previous study (7), in

cases where TRNs are available (62–67). Another alternative is inferring TRNs from gene expression data (23, 24, 68). The latter methods have also been used in elucidating the biological function of different regulons, providing an additional functional layer to the TRN (23, 24, 69). Such analyses could allow studying common strategies involving transcriptional adaptation.

**Correlated changes within a subgroup of genes are associated with mucoid strains.** The clustering analysis allowed us to demarcate a subset of progenitor-progeny pairs utilizing a coherent adaptive strategy, embodied by the identical changes in the predicted expression of a subgroup of genes. This subset, corresponding to cluster 14, was strongly enriched with mucoid isolates, a major adaptive phenotype found in *P. aeruginosa* chronic infections. We hypothesize that the changes in these genes constitute a transcriptional arm of the mucoid adaptive strategy. Other clusters may correspond to similar adaptive strategies, yet to be identified (see Text S1 in the supplemental material for a discussion of the changes in cluster 13). Identification and definition of different adaptive strategies utilized by pathogens during chronic infection have major promise in the treatment of persistent infections and open the door to personalized tailored antibiotic treatment in the future. Comprehending the selective pressures leading to those different strategies could allow us to drive the evolution of pathogens during infections toward strategies employing lesser virulence or ones that are easier to treat. We believe future research on this subject should focus on distinguishing and clarifying the different adaptive strategies, assessing their association with clinical outcomes and their susceptibility to different antimicrobial agents.

**Possible interplay between adaptation at the genetic and transcriptional levels.** We postulated that processes associated with transcriptional adaptation may not be clearly apparent at the level of single genes, leading us to expand our study to KEGG pathways and GO annotations. We indeed detected additional processes that were not identified in previous analyses and were not previously linked to host adaptation. We also reassociated LPS with host adaptation through the predicted consistent decreased synthesis of its components, a completely different mechanism than previously demonstrated. The possible interplay between mutations in genes taking part in the biosynthesis of LPS (7) and the decreased expression of genes synthesizing L-rhamnose is intriguing. In addition to LPS, we found substantial overlap between pathways and annotations with high HS and low mROS values and pathways and annotations that we previously found to accumulate missense mutations. The interpretation of missense mutations, leading to changes in amino acid sequences during infection, has remained elusive (7). Our current results indicate that pathways with genes accumulating such mutations are also subjected to selection for decreased expression. This supports the interpretation that such mutations lead to impaired function of the genes and act in concert with the transcriptional adaptive strategies. Our findings further expand the concept of convergent adaptation, with different strains possibly achieving similar effects on the activity of a pathway via both genetic and transcriptional strategies. Importantly, our results are in line with previous studies including small numbers of patients. Camus et al. (3) recently reviewed studies comparing gene expression between early and late CF *P. aeruginosa* isolates and identified several genes that showed expression change in multiple isolates across the different studies. The *P. aeruginosa* TRN includes 21 of these genes, and we predict 16 of those to have extreme mROS values in the same direction as that noted by Camus et al. (3).

Bacterial pathogens use miscellaneous mechanisms to adapt to the host environment during acute and chronic infection, including changes in gene expression and genetic mutations. Our study has revealed novel interplay between these two levels. The effect of some genetic mutations becomes apparent only at the transcriptional level, and different strains can achieve the same adaptive strategies via either the genetic level or propagation through the TRN. This interplay demonstrates the versatility of the adaptive arsenal of bacterial pathogens and their ability to adapt to the host conditions in a myriad of ways.

## MATERIALS AND METHODS

***P. aeruginosa* transcriptional regulatory network.** We reconstructed the *P. aeruginosa* PAO1 TRN from published data compiled by Galán-Vásquez et al. (25) using Python's *networkx* package (70). We assigned each edge in the TRN a weight according to its regulation type. Edges representing upregulation were assigned a weight of $+1$, while downregulation edges were assigned a weight of $-1$.

***P. aeruginosa* clinical isolates.** We utilized sequencing data from 114 *P. aeruginosa* strains from different patients, sequenced at multiple time points during chronic infections, previously curated from the literature (7). While the experimental methods for culturing and sequencing the samples differed between various studies, all samples were derived from sputum samples, except for two strains whose samples were derived from endotracheal and bronchial aspirate, respectively (71, 72). Isolates were clonally purified from clinical specimens and were mostly sequenced by short-read sequencing, except for the study by Wang et al. (32), in which the isolates underwent sequencing by long reads using PacBio RS II. We extracted progenitor-progeny isolate pairs and genes undergoing LOF for each such pair from our previous analysis (7).

**Computing the regulatory outcome score (ROS).** We extracted two gene lists from the data of Gatt and Margalit (7) for each progenitor-progeny isolate pair: (a) genes present in the progenitor isolate and (b) genes predicted to undergo LOF between the progenitor and progeny isolates.

We recreated the *P. aeruginosa* PAO1 TRN from the work of Galán-Vásquez et al. (25) for each progenitor-progeny pair using only the genes appearing in the progenitor isolate. For each gene we computed its regulatory outcome score (ROS), which is the predicted change in the expression of the gene between the progenitor and progeny isolates resulting from all relevant LOF events. ROS values were computed as follows (Fig. 7).

We estimated the effect of a TF/SF-encoding gene undergoing LOF on each gene downstream of it in the TRN by considering the number of edges in the shortest path between the two genes (path length) and the product of the weights of those edges. The path could not include other genes undergoing LOF themselves. For example, assuming gene $X$ encoding a TF underwent LOF between the progenitor and progeny isolates, we computed its effect on gene $Y$ located downstream of $X$ in the TRN [$\text{score}(Y_X)$], by multiplying the weights of the edges ($e$) from $X$ to $Y$ $\left[\prod_{e \in \text{path}(X,Y)} \text{weight}(e)\right]$, dividing the product by the number of edges between them [path length $(X,Y)$], and negating the result (as gene $X$ undergoes LOF, the predicted effect is in the opposite direction from the effect exerted by $X$).

$$\text{score}(Y_X) = -\frac{\prod_{e \in \text{path}(X,Y)} \text{weight}(e)}{\text{path length } (X, Y)}$$

We calculated the sum of the scores each gene obtained by the effects of all its upstream TF/SF-encoding genes undergoing LOF, under the assumption that TFs/SFs affect the gene in an additive manner. The final ROS assigned to a gene per a progenitor-progeny pair is 1 if the sum is positive, $-1$ if the sum is negative, and 0 if the sum is exactly 0. For example, gene $Y$'s score, affected by the genes $X$ and $Z$ that undergo LOF upstream from it, would be calculated as follows:

$$\text{ROS}(Y) = \begin{cases} -1, & \text{score}(Y_X) + \text{score}(Y_Z) < 0 \\ 0, & \text{score}(Y_X) + \text{score}(Y_Z) = 0 \\ +1, & \text{score}(Y_X) + \text{score}(Y_Z) > 0 \end{cases}$$

By applying this computation to each gene, we obtain the ROS for every gene in the network for each progenitor-progeny pair. Genes undergoing LOF themselves were not assigned an ROS for the progenitor-progeny pair. The mean ROS (mROS) of each gene is the mean of the ROS values it obtains in the networks of all the progenitor-progeny pairs where the gene is found in the progenitor isolate and does not undergo LOF itself.

Gene content differs between different *P. aeruginosa* strains, and rare genes may appear in only a small number of progenitor-progeny isolate pairs. The mROS of such genes may therefore be biased by rare LOF events randomly occurring in specific pairs. To avoid this possible bias, we filtered out genes appearing in fewer than 10 progenitors.

**Assessing the relation between ROS and the time interval between the sampling of the progenitor and progeny isolates.** We tested whether the time interval between the sampling of the progenitor and progeny isolates in each pair has an impact on the ROS. For each gene we computed the correlation between the ROS value and the interval between the sampling times of the isolates in a progenitor-progeny pair, across the different progenitor-progeny pairs. All *P* values were corrected for the testing of multiple hypotheses using the Benjamini-Hochberg method (42).

**Statistical framework to assess mROS.** We developed a statistical method to assess whether the mROS of a gene differed statistically significantly from random expectation. We counted how many genes underwent LOF for each progenitor-progeny pair and ran 100,000 simulations, randomly choosing the same number of genes to undergo LOF in each one. For each gene, we then assessed the fraction of simulations where the absolute value of the mROS was equal to or greater than the absolute value of the actual mROS of the gene in our data. We treated this fraction as the *P* value for the gene and corrected all *P* values for multiple hypothesis testing using the Benjamini-Hochberg method (42). We considered genes with corrected *P* values of ≤0.1 as having an mROS that statistically significantly differs from random expectation.

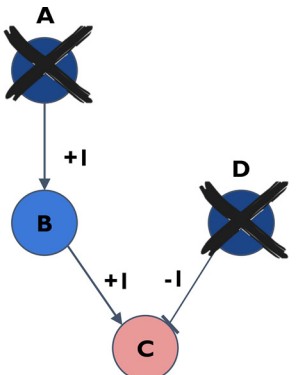

$$distance(A, C) = 2$$
$$edges(A, C) = 1 \cdot 1$$
$$Score\ C_A = -\frac{1 \cdot 1}{2} = -0.5$$

$$distance(D, C) = 1$$
$$edges(D, C) = -1$$
$$Score\ C_D = -\frac{-1}{1} = 1$$

$$Score\ C = \begin{cases} -1, & C_A + C_D < 0 \\ 0, & C_A + C_D = 0 = \\ +1, & C_A + C_D > 0 \end{cases} \begin{cases} -1, & -0.5 + 1 < 0 \\ 0, & -0.5 + 1 = 0 = +1 \\ +1, & -0.5 + 1 > 0 \end{cases}$$

**FIG 7** TRN propagation algorithm. An example of the computation of the regulatory outcome score (ROS) by the TRN propagation algorithm. Gene C (node C) is positively regulated by TF B (node B) and negatively regulated by TF D (node D). TF B is positively regulated by TF A (node A). In the progenitor-progeny pair in the example, TFs A and D were determined to undergo loss-of-function (LOF). In the blue box, the effect of the LOF of TF A on gene C is demonstrated. TF A is two edges away from gene C, and the product of the weights of all the edges between them is 1 × 1 = 1. The final score of the impact of TF A on gene C is therefore −1 × (product of weights)/number of edges = −0.5. In the red box, the effect of the LOF of TF D on gene C is demonstrated. TF D is one edge away from gene C, and the product of the weights of all the edges between them is −1. The final score of TF D on gene C is therefore −1 × (product of weights)/number of edges = 1. In the green box, the final ROS of gene C is demonstrated. It is defined as the sign of the sum of the scores of gene C corresponding to all the TFs predicted to affect its expression in the progenitor-progeny pair, in this case the sign of −0.5 + 1, which is +1. The expression of gene C is therefore predicted to increase in the progenitor-progeny pair due to the summed effects of all LOF events.

**Computing the heterogeneity score.** We defined a heterogeneity score (HS) for each gene in the network, representing the variety of paths leading to change in expression of that gene. To calculate it, for each gene, we examined its ROS in the different progenitor-progeny pairs and the TF/SF-encoding genes undergoing LOF involved in the calculation of that ROS. We calculated the HS of each gene in the network as shown below.

1. Within each progenitor-progeny pair where the sign of the ROS is equal to the sign of the gene's mROS, we extracted TFs/SFs whose LOF was predicted to affect the gene in the same direction as the sign of its ROS in that progenitor-progeny pair.
2. We combined all the lists for the different progenitor-progeny pairs and counted the number of times each TF/SF appeared. We divided the result by the path length between the gene and the TF/SF. For example, the score of gene Y with respect to TF/SF X is computed as follows:

$$score(Y_X) = \frac{\text{number of LOF events}(X)}{\text{path length}(X, Y)}$$

3. We normalized the computed score of gene Y with respect to X by dividing it by the maximal score Y obtained in respect to any TF/SF affecting it. Therefore, the normalized score of gene Y concerning TF/SF X is

$$\text{normalized score}(Y_X) = \frac{score(Y_X)}{\max(\text{score of } Y)}$$

4. The final HS of a gene is computed as the sum of all its normalized scores.

This calculation emphasizes the effect of close TFs/SFs undergoing LOF in a large number of progenitor-progeny pairs on the HS. Thus, the score is an indicator of the number of LOF events of the same magnitude that affected the gene of interest.

**Association between mROS and HS and adaptive KEGG pathways and GO annotations.** We studied the association between mROS/HS and KEGG pathways determined as adaptive in our previous study (7) using two methods. (i) We used the Student *t* test to compare the mROS of genes associated with adaptive KEGG pathways with the mROS of all other genes. We performed a similar test for HS. (ii) We performed Fisher's exact test to assess enrichment of adaptive KEGG pathways with pathways with statistically significantly high mROS and HS values, i.e., we assessed whether there are more adaptive KEGG pathways with statistically significant mROS values than expected at random based on the number of adaptive pathways and the number of pathways with statistically significant mROS values. We performed similar analyses for GO annotations.

**Gene clustering analysis.** We first constructed a vector for each gene including its ROS values across all progenitor-progeny pairs. We then calculated the Spearman correlation coefficient (*r*) between each two vectors, considering only progenitor-progeny pairs where both genes were found in the progenitor isolate. Since different genes that are functionally associated can change in opposite directions (i.e., up- and

downregulation), we used $1 - |r|$ as the distance measure for the clustering. We removed genes with ROS values of 0 across all strains, as well as genes appearing in fewer than 10 progenitor isolates. Hierarchical clustering was performed using the Python SciPy package. The cutoff point of the dendrogram for the different clusters was determined as 0.6 to maximize the average silhouette score (73) without losing pairs of genes that are strongly correlated and do not share transcriptional regulators (which are functionally intriguing).

**Calculating the concordance score of a progenitor-progeny pair in respect to a cluster.** In order to determine whether the predicted expression change of a gene in each progenitor-progeny pair is in concordance with its mROS, we multiplied the ROS of the gene within each pair by $-1$ if the gene's mROS was negative or by 0 if the gene's mROS was 0, or we used the ROS as is if the gene's mROS was positive, obtaining a score of $0/-1/+1$ for each gene relevant to a progenitor-progeny pair. We then averaged the scores of all genes in each cluster for each progenitor-progeny pair, obtaining a concordance score for that pair for the cluster.

**Enrichment of *mucA* in progenitor-progeny pairs with higher concordance scores for cluster 14.** We curated high-impact *mucA* mutations from snpEff results reported in our previous study (7). We calculated the concordance scores of progenitor-progeny pairs with *mucA* mutations for cluster 14 and compared them with the concordance scores of other progenitor-progeny pairs for cluster 14 using a one-tailed Mann-Whitney U test.

**Data and script availability.** Accession numbers for all relevant sequencing data are listed in Table S1 of our previous study (7). The scripts used for the analysis of the data are available through the following GitHub repository: https://github.com/danasav/Convergent-within-host-adaptation-of-Pseudomonas-aeruginosa-through -the-TRN.

## SUPPLEMENTAL MATERIAL

Supplemental material is available online only.
**TEXT S1**, DOCX file, 0.1 MB.
**FIG S1**, TIF file, 0.4 MB.
**FIG S2**, TIF file, 0.8 MB.
**FIG S3**, TIF file, 0.5 MB.
**FIG S4**, TIF file, 4.6 MB.
**FIG S5**, TIF file, 1.2 MB.
**TABLE S1**, XLSX file, 0.02 MB.
**TABLE S2**, XLSX file, 0.4 MB.
**TABLE S3**, XLSX file, 0.1 MB.
**TABLE S4**, XLSX file, 0.1 MB.

## ACKNOWLEDGMENTS

We are grateful to I. Rosenshine and R. Hershberg for their support and comments. We thank E. Galán-Vásquez for his efficient and helpful cooperation.

This work was supported by the Israel Science Foundation administered by the Israeli Academy for Sciences and Humanities (grant 876/17). Y.E.G. is partially supported by fellowships of the Foulkes Foundation and the Data Sciences program of the Planning and Budgeting Committee of the Israel Council for Higher Education.

All authors contributed extensively to the work presented in this paper. H.M. and Y.E.G. conceived the study, Y.E.G., T.B., and D.S. analyzed the data, Y.E.G. and H.M. supervised the work and wrote the manuscript.

We declare no competing interests.

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
