## [Reviewer comments · mSystems]

Convergent within-host adaptation of *Pseudomonas aeruginosa* through the transcriptional regulatory network

Yair Gatt, Dana Savion, Tal Bamberger, and Hanah Margalit

Corresponding Author(s): Hanah Margalit, Hebrew University of Jerusalem

Review Timeline:

Submission Date:	January 11, 2023
Editorial Decision:	February 8, 2023
Revision Received:	February 14, 2023
Accepted:	March 3, 2023

Editor: Mani Arumugam

Reviewer(s): Disclosure of reviewer identity is with reference to reviewer comments included in decision letter(s). The following individuals involved in review of your submission have agreed to reveal their identity: Janne Gesine Thöming (Reviewer #2)

Transaction Report:

DOI: <https://doi.org/10.1128/msystems.00024-23>

February 8, 2023

Prof. Hanah Margalit
Hebrew University of Jerusalem
Microbiology and Molecular Genetics
IMRIC
Faculty of Medicine
Jerusalem 9112102
Israel

Re: mSystems00024-23 (Convergent within-host adaptation of *Pseudomonas aeruginosa* through the transcriptional regulatory network)

Dear Prof. Hanah Margalit:

Thank you for submitting your manuscript to mSystems. We have completed our review and I am pleased to inform you that, in principle, we expect to accept it for publication in mSystems. However, acceptance will not be final until you have adequately addressed the minor reviewer comments. Please find them below.

Preparing Revision Guidelines

Sincerely,

Mani Arumugam

Editor, mSystems

Journals Department
American Society for Microbiology

Reviewer comments:

Reviewer #2 (Comments for the Author):

I would like to thank the authors for their thorough response to the raised concerns. My points were answered in detail, both in the rebuttal letter and in the newly added and revised text passages in the manuscript. Although the approach presented is quite unique, I think the analysis is now much easier to follow. The significance and relevance of the gene expression changes predicted in this work are clearly supported by the integration of additional phenotypic data, e.g. with regards to antibiotic susceptibility/resistance. Most ambiguities have been resolved. The manuscript presents interesting and important insights into the role of convergent evolution and the interplay of (patho-)adaptive mutations and their transcriptional consequences.

I have only a few minor comments/suggestions:

P. 3, line 28: Wording - maybe "patients with CF" instead of "CF"

P. 3 line 30: In the context of evolution of tolerance and the subsequent evolution of resistance, the work of Levin-Reisman et al (<https://doi.org/10.1126/science.aaj2191>) and Santi et al (<https://doi.org/10.1128/mBio.03482-20>) would fit in quite nicely.

P. 5 line 20: Do I understand correctly that of the 163 nodes representing TFs/SFs in the TRN used, only the 29 TFs/SFs shown in Fig. S1b have LOF mutations in more than one progenitor-progeny pair? I think this number should be mentioned in the results section. The finding that LOF mutations occur in <20% of the included TFs/SFs is not uninteresting.

P. 8 line 12: Have you checked whether (LOF-) mutations in the repressor/negative regulator mexR/PA0242 (e.g. <https://doi.org/10.1073/pnas.0805489105>) could perhaps be the genetic correlate for increased mexAB-oprM expression? As indicated in Fig. S1b, these mutations occurred in some of the progenitor-progeny pairs.

P. 14 line 2: Wording - "Predicted convergent transcriptional adaptation..."

P. 16 line 6: Wording - "..., embodied by the identical changes the predicted expression..."

Fig. 5: The resolution of the writing could probably be improved (except for "Polyketide sugar unit biosynthesis" which seems to be manually edited)

Fig. 6b: Could you please add the number of datapoints for both groups in the figure caption (pairs with and without mucA LOF mutations).

Supplementary Results:

Page 3: Predicted reduced synthesis of HCN is quite an interesting finding, since hcn genes are, among others, regulated by the quorum sensing master regulators RhIR and LasR (hotspot for patho-adaptive mutations). I think the possible link to QS/mutations in LasR could be mentioned here.

Page 4: Could you please provide a reference that beta lactamases are secreted by the MexAB-OprM efflux pump? To my knowledge, beta lactamases are released either by cell lysis or by outer membrane vesicles.

The Hebrew University of Jerusalem
Faculty of Medicine, IMRIC
Department of Microbiology and Molecular Genetics
Tel. 972-2-6758614, 6758647
Fax 972-2-6757308
e-mail: hanahm@ekmd.huji.ac.il

February 12th, 2023

Dear Prof. Arumugam,

Below please find a detailed list of the revisions made.

With best regards,

Hanah Margalit

List of revisions (Please note that the line numbering we refer to is the one adjacent to the text)

Reviewer 2

I would like to thank the authors for their thorough response to the raised concerns. My points were answered in detail, both in the rebuttal letter and in the newly added and revised text passages in the manuscript. Although the approach presented is quite unique, I think the analysis is now much easier to follow. The significance and relevance of the gene expression changes predicted in this work are clearly supported by the integration of additional phenotypic data, e.g. with regards to antibiotic susceptibility/resistance. Most ambiguities have been resolved. The manuscript presents interesting and important insights into the role of convergent evolution and the interplay of (patho-)adaptive mutations and their transcriptional consequences.

I have only a few minor comments/suggestions:

P. 3, line 28: Wording - maybe "patients with CF" instead of "CF"

Done (page 3, line 28).

P. 3 line 30: In the context of evolution of tolerance and the subsequent evolution of resistance, the work of Levin-Reisman et al (<https://doi.org/10.1126/science.aaj2191>) and Santi et al (<https://doi.org/10.1128/mBio.03482-20>) would fit in quite nicely.

We have added references to these papers (page 3, line 30).

P. 5 line 20: Do I understand correctly that of the 163 nodes representing TFs/SFs in the TRN used, only the 29 TFs/SFs shown in Fig. S1b have LOF mutations in more than one progenitor-progeny pair? I think this number should be mentioned in the results section. The finding that LOF mutations occur in <20% of the included TFs/SFs is not uninteresting.

We added these results to the manuscript (page 5, lines 26-28).

P. 8 line 12: Have you checked whether (LOF-) mutations in the repressor/negative regulator mexR/PA0242 (e.g. <https://doi.org/10.1073/pnas.0805489105>) could perhaps be the genetic correlate for increased mexAB-oprM expression? As indicated in Fig. S1b, these mutations occurred in some of the progenitor-progeny pairs.

The *mexR* gene did not undergo an LOF mutation in this patient. However, *mexR* is predicted to have reduced expression via the TRN in the relevant progenitor-progeny pair. This possibly explains the predicted increased expression of *mexAB-oprM*. We have added this information to the manuscript (page 8, line 13).

P. 14 line 2: Wording - "Predicted convergent transcriptional adaptation..."

Done (page 14, line 2).

P. 16 line 6: Wording - "..., embodied by the identical changes the predicted expression..."

Done (page 16, line 6).

Fig. 5: The resolution of the writing could probably be improved (except for "Polyketide sugar unit biosynthesis" which seems to be manually edited)

We improved the resolution of this figure.

Fig. 6b: Could you please add the number of datapoints for both groups in the figure caption (pairs with and without muca LOF mutations).

We have added the number of data points, there are 28 progenitor-progeny pairs with *muca* LOF mutations (Fig. S1b, Table S2) and 158 pairs without such mutations. Note that the figure was slightly changed. The previous version did not exclude progenitor-progeny pairs

with preexisting *mucA* LOF mutations. After excluding these pairs from the analysis, the difference between the groups is even more pronounced than before.

Supplementary Results:

Page 3: Predicted reduced synthesis of HCN is quite an interesting finding, since hcn genes are, among others, regulated by the quorum sensing master regulators RhlR and LasR (hotspot for patho-adaptive mutations). I think the possible link to QS/mutations in LasR could be mentioned here.

We addressed this possible link (Supplementary Information page 3, lines 11-15). Notably, the Heterogeneity scores of all genes in this pathway are very high, meaning their predicted changes in expression are the result of different genetic events in different progenitor-progeny pairs. Therefore, even if the predicted changes in some pairs are linked to LOF mutations in *lasR*, they could not explain the changes across all pairs.

Page 4: Could you please provide a reference that beta lactamases are secreted by the MexAB-OprM efflux pump? To my knowledge, beta lactamases are released either by cell lysis or by outer membrane vesicles.

The review we cited by Poole addresses this subject (1). According to the review, the MexAB-OprM efflux pump accommodates the broadest range of β -Lactams, including carbapenems, penicillin-ticarcillin, and aztreonam. We added some of the references cited by Poole on this subject to strengthen the connection (2-5).

1. Poole K. 2011. *Pseudomonas aeruginosa*: resistance to the max. *Frontiers in Microbiology* 2:65.
2. Drissi M, Ahmed ZB, Dehecq B, Bakour R, Plésiat P, Hocquet D. 2008. Antibiotic susceptibility and mechanisms of β -lactam resistance among clinical strains of *Pseudomonas aeruginosa*: First report in Algeria. *Médecine et Maladies Infectieuses* 38:187-191.
3. Okamoto K, Gotoh N, Nishino T. 2002. Alterations of susceptibility of *Pseudomonas aeruginosa* by overproduction of multidrug efflux systems, MexAB-OprM, MexCD-OprJ, and MexXY/OprM to carbapenems: Substrate specificities of the efflux systems. *Journal of Infection and Chemotherapy* 8:371-373.
4. Pai H, Kim J, Kim J, Lee JH, Choe KW, Gotoh N. 2001. Carbapenem resistance mechanisms in *Pseudomonas aeruginosa* clinical isolates. *Antimicrob Agents Chemother* 45:480-484.
5. Tomás M, Doumith M, Warner M, Turton JF, Beceiro A, Bou G, Livermore DM, Woodford N. 2010. Efflux Pumps, OprD Porin, AmpC β -Lactamase, and Multiresistance in *Pseudomonas aeruginosa* Isolates from Cystic Fibrosis Patients. *Antimicrobial Agents and Chemotherapy* 54:2219-2224.

March 3, 2023

Prof. Hanah Margalit
Hebrew University of Jerusalem
Microbiology and Molecular Genetics
IMRIC
Faculty of Medicine
Jerusalem 9112102
Israel

Re: mSystems00024-23R1 (Convergent within-host adaptation of *Pseudomonas aeruginosa* through the transcriptional regulatory network)

Dear Prof. Hanah Margalit:

Your manuscript has been accepted, and I am forwarding it to the ASM Journals Department for publication. For your reference, ASM Journals' address is given below. Before it can be scheduled for publication, your manuscript will be checked by the mSystems production staff to make sure that all elements meet the technical requirements for publication. They will contact you if anything needs to be revised before copyediting and production can begin. Otherwise, you will be notified when your proofs are ready to be viewed.

If you would like to submit a potential Featured Image, please email a file and a short legend to msystems@asmusa.org. Please note that we can only consider images that (i) the authors created or own and (ii) have not been previously published. By submitting, you agree that the image can be used under the same terms as the published article. File requirements: square dimensions (4" x 4"), 300 dpi resolution, RGB colorspace, TIF file format.

We recognize that the video files can become quite large, and so to avoid quality loss ASM suggests sending the video file via <https://www.wetransfer.com/>. When you have a final version of the video and the still ready to share, please send it to mSystems staff at msystems@asmusa.org.

Sincerely,

Mani Arumugam
Editor, mSystems

Journals Department
E-mail: mSystems@asmusa.org